# Real-time Classification, Geolocation and Interactive Visualization of COVID-19 Information Shared on Social Media to Better Understand Global Developments

**Andrei Mircea**
MILA/McGill University
Montreal, QC, Canada
`mirceara@mila.quebec`

## Abstract

As people communicate on social media during COVID-19, it can be an invaluable source of useful and up-to-date information. However, the large volume and noise-to-signal ratio of social media can make this impractical. We present a prototype dashboard for the real-time classification, geolocation and interactive visualization of COVID-19 tweets that addresses these issues. We also describe a novel L2 classification layer that outperforms linear layers on a dataset of respiratory virus tweets.

## 1 Introduction

As the COVID-19 pandemic continues to rapidly evolve on a global scale, members of the public and crisis managers alike need access to digestible, useful, and up-to-date information. Such information is crucial in enabling informed and responsive decision making that minimizes risk and mitigates harm caused by the virus. Throughout crises, social media platforms in particular have been invaluable to crisis managers as they provide unparalleled situational awareness through grassroots citizens reporting (Burns and Shanley, 2012). Furthermore, various organizations and individuals share the most recent and relevant information on social media: enabling affected people to stay informed, and crisis managers to measure outreach effectiveness (Jin et al., 2014).

However, just as social media can empower the effective dissemination of useful information, it also enables the spread of harmful misinformation. In fact, the signal-to-noise ratio of information shared on social media is a well-known problem (Imran et al., 2016), and misinformation has been a pervasive problem throughout the COVID-19 pandemic (Brennen et al., 2020). Extracting up-to-date, useful, and digestible information from social media platforms also has its own challenges:

- **Digestible**: the sheer volume of posts shared on social media can make it an impractical source of information (e.g. Banda et al. (2020) report 4 million COVID-19 tweets per day);

- **Useful**: different people in different places need different information, what is useful to a student in Montreal may not be useful to a crisis manager in Beijing;

- **Up-to-date**: due to the large velocity of social media posts and the speed at which a situation can evolve, the most up-to-date information may rapidly change.

To address these challenges, we built a prototype dashboard that classifies and geolocates COVID-19 tweets in real-time, making it easier to explore useful and up-to-date information in a digestible and intuitive way (Figure 1). The primary components of our system are:

- **Text-based geolocation of tweets**: Filtering information by location is an important first step in ensuring its usefulness and digestibility. Unfortunately, less than 1% of tweets are geotagged (Marciniec, 2019). We use named entity-recognition and a gazetteer to address this issue;

- **L2 classification of tweets**: Categorizing tweets can also improve their usefulness and digestibility. We fine-tune BERT on respiratory virus tweet classification using a novel metric-learning inspired L2 classification layer that outperforms linear layers;

- **Interactive map**: Our tweet geolocation and classification allows us to organize and present useful information on an interactive map that emphasizes the most up-to-date tweets and makes it easier to navigate the large volume of information in an intuitive way.

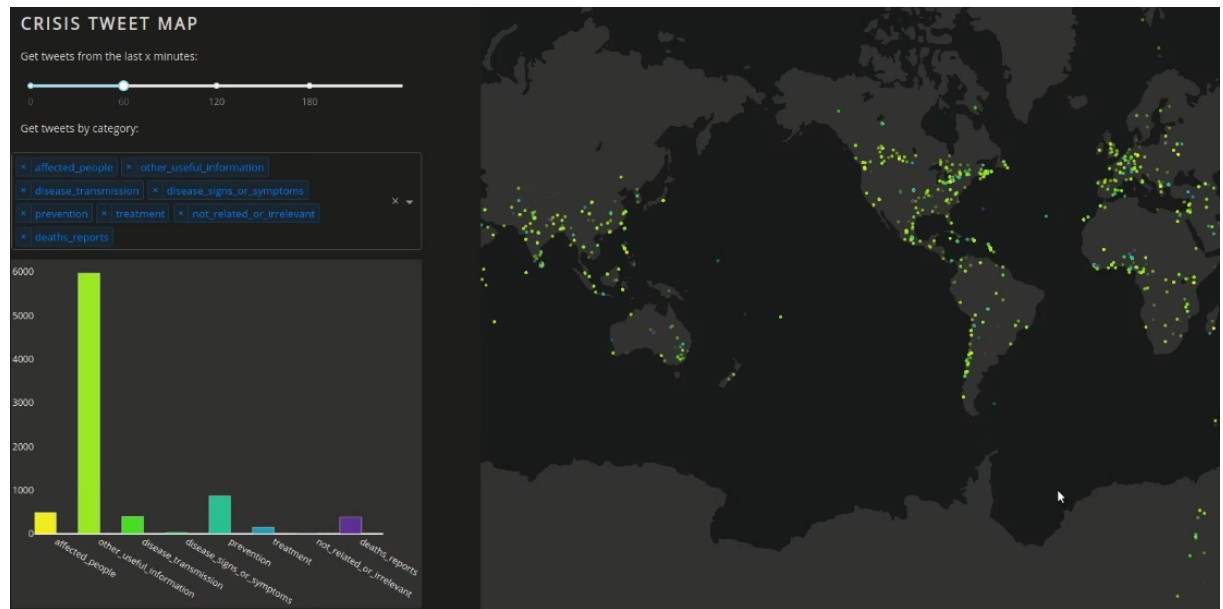

Figure 1: screenshot of dashboard showcasing global scale of COVID-19 information shared on social media.

| Class | Description |
|---|---|
| Disease signs or symptoms | Reports of symptoms such as fever, cough, diarrhea, and shortness of breath or questions related to these symptoms |
| Disease transmission | Reports of disease transmission or questions related to disease transmission |
| Disease Prevention | Questions or suggestions related to the prevention of disease or mention of a new prevention strategy |
| Disease Treatment | Questions or suggestions regarding the treatments of the disease |
| Death reports | Reports of deaths due to the disease |
| Affected people | Reports of affected people due to the disease |
| Other useful information | Other useful information that is related to the disease |
| Not related or irrelevant | Unrelated to the situation or irrelevant |

Table 1: Class descriptions provided to annotators of the MERS 2014 tweet dataset.

## 2 System Overview

### 2.1 Tweet Streaming

To collect relevant tweets in real-time, we used the Twitter `statuses/filter` API with the following search terms: `coronavirus`, `covid19`, `covid-19`, `covid`, filtering english tweets. In Banda et al. (2020), over 80% of COVID-19 tweets are retweets (i.e. the same content shared by different people), therefore we also filter out retweets to reduce the volume of information and make it more digestible.

### 2.2 Tweet Classification

#### 2.2.1 Training Data

To automatically classify COVID-19 tweets into useful categories, we fine-tune a pooled `bert-base-cased` pretrained language model (Devlin et al., 2018) on a dataset of labeled tweets from the 2014 Middle-Eastern Respiratory Syndrome (MERS) outbreak (Imran et al., 2016). Class descriptions are shown in Table 1 while class counts are shown in Table 2. The dataset was split into a stratified train-test split. To address the issue of class imbalance during training, we oversample elements of every class until the same number of elements as the majority class is reached.

#### 2.2.2 L2 Classification

Traditional linear classification layers that map $D$-dimensional embeddings to $N$ logits via a $D \times N$ matrix multiplication are performing metric learning in the inner product space (Chen et al., 2018). However, inner product similarity has a bias against rare classes which have low-norm embeddings (Demeter et al., 2020) and prevents the use of learned representations in downstream tasks which use L2 distance, e.g. K-means clustering. We propose a novel classification layer that computes negative L2 distances as logits instead. The latency of our model is comparable to that of a traditional linear classification layer, observing inference speeds of 400-750ms on a consumer-grade machine with a 2080Ti GPU and an i5-8600K CPU.

Our approach obtains a best validation accuracy of 76.84%, compared to 76.35% for the traditional inner-product based method after 20 epochs of training with a batch size of 16, an adamw optimizer, and a learning rate of $1e-5$, taking evaluation metrics from the epoch with the best validation loss. Despite the limited increase in accuracy, we find that our approach significantly improves the F1 score of rare classes suggesting it addresses the bias of inner product similarity (Table 2).

| Class | Count | | F1 | |
|---|---|---|---|---|
| | Train | Test | IP | L2 |
| affected_people | 559 | 140 | 68.36 | 72.71 |
| other_useful_information | 554 | 139 | 66.56 | 75.50 |
| disease_transmission | 199 | 50 | 61.43 | 76.04 |
| disease_signs_or_symptoms | 124 | 32 | 73.32 | 74.12 |
| prevention | 69 | 18 | 71.57 | 79.63 |
| treatment | 60 | 15 | 77.63 | 86.15 |
| not_related_or_irrelevant | 23 | 6 | 73.20 | 84.46 |
| deaths_reports | 23 | 6 | 84.66 | 87.64 |

Table 2: Class counts and per-class validation % F1 scores for IP (inner product) and L2 classification layers on the MERS 2014 dataset.

Following reviewer feedback, we also evaluate our approach on tweets from the 2014 Pakistan Floods (Imran et al., 2016) to more rigorously verify this hypothesis. However we find our approach does not consistently improve F1 scores (Table 3), indicating there are other factors at play beyond the bias of inner product similarity.

| Class | Count | | F1 | |
|---|---|---|---|---|
| | Train | Test | IP | L2 |
| other_useful_information | 558 | 140 | 52.08 | 47.84 |
| donation_needs_or_offers_or_volunteering | 423 | 106 | 70.80 | 66.56 |
| injured_or_dead_people | 207 | 52 | 78.82 | 78.28 |
| sympathy_and_emotional_support | 101 | 26 | 76.81 | 72.39 |
| missing_trapped_or_found_people | 93 | 24 | 53.17 | 56.98 |
| displaced_people_and_evacuations | 84 | 22 | 62.76 | 59.63 |
| infrastructure_and_utilities_damage | 75 | 19 | 71.24 | 73.05 |
| caution_and_advice | 44 | 12 | 77.74 | 72.31 |
| not_related_or_irrelevant | 21 | 6 | 76.21 | 73.85 |

Table 3: Class counts and per-class validation % F1 scores for IP (inner product) and L2 classification layers on the 2014 Pakistan Flood dataset.

### 2.2.3 Text-based geolocation

To geolocate tweets from their text content, we use the named entity recognizer from Honnibal and Montani (2017) to extract name places, then match them to the GeoNames gazetteer to extract geo-coordinates (Halterman, 2017). We divide geolocations into those extracted from user-provided tweet geo-tags, as well as cities and countries extracted from tweet texts and from user bio texts. We then allow users to filter their preferred types of geolocation. Tweets with multiple geolocations (e.g. a Canadian in the UK tweeting about France) are duplicated on the map.

### 2.2.4 Visualization

Classified and geolocated tweets are mapped in real-time. While the geolocation allows users to intuitively navigate geographically relevant content, tweets are also color-coded along their different categories to make it easier for users to find tweets relevant to them. The opacity of a tweet is also proportional to its recency, such that more recent tweets pop out on the map and allow users to easily track the most up-to-date information. Lastly, the dashboard makes it easy to filter tweets along different categories, timespans or keywords, and to visualize a tweet's text by hovering over it.

## 3 Related Work

The real-time visualization of classified and geolocated tweets on an interactive map has been demonstrated before. For example, Sankaranarayanan et al. (2009) use a Naive Bayes classifier to identify tweets about news, and statistical NER combined with a gazetteer to geoparse tweets by extracting and resolving location names (toponyms) from their content.

Middleton et al. (2014) apply a similar geoparsing approach to tweets from natural disasters, as they find less than 1% of tweets are geotagged. However, due to the highly local nature of natural disasters, they eschew NER in favor of simple string matching with a predetermined set of valid toponyms. Furthermore, they do not perform tweet classification.

In contrast, Benitez et al. (2018) classify tweets by disaster type, but do not perform geoparsing, relying on geotagged tweets instead. Similar other works classify, geoparse, and map tweets in real-time during various crises (Choi and Bae, 2015; Mao et al., 2018; Avvenuti et al., 2018; Anbalagan and Valliyammai, 2016; Hernandez-Suarez et al., 2019).

To the best of our knowledge, we are the first to combine, in a real-time tweet mapping interface, both neural NER geoparsing and multi-class classification for a single event type of global scale.

## 4 Conclusion

We present a prototype dashboard for the real-time classification, geolocation and interactive visualization of COVID-19 information shared on social media. We design our system to make it as easy as possible for users to find useful and up-to-date information in a digestible format, addressing fundamental issues in extracting valuable information from social media during crises.

More specifically, we leverage natural language processing to geolocate a portion of the large amount of non-geotagged tweets we capture, helping users find geographically relevant information. We also fine-tune a novel L2 classification layer to categorize tweets in useful labels, helping users structure and navigate the large volume of tweets we capture.

Lastly, we also show our novel L2 classification layer surpasses traditional linear layers in terms of accuracy and per-class F1 scores on a dataset of respiratory virus tweets. However, following reviewer feedback we find that this improvement is not consistent on other datasets, suggesting the need for future research to better understand the factors at play.

## 5 Future Work

Our dashboard prototype has several opportunities for improvement. These make interesting potential avenues for future research in natural language processing and include:

- Automatic term seeding for capturing a greater proportion of relevant tweets through the Twitter API;

- Better geoparsing to capture provinces and reduce false positives from place-like names;

- Unsupervised or human-in-the-loop tweet-clustering to dynamically capture novel fine grained topics not present in past labeled data;

- Identification of misinformation and first-person narratives in tweets to better assist crisis managers and affected people;

## Acknowledgments

We'd like to thank Renee Sieber for her insights on applying NLP to crisis management, and the reviewers for their valuable feedback.

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
