# OpenReview forum: "Real-time Classification, Geolocation and Interactive Visualization of COVID-19 Information Shared on Social Media to Better Understand Global Developments"
_EMNLP/2020/Workshop/NLP-COVID — NLP-COVID19-EMNLP Poster_

### Official Review · AnonReviewer3 · 2020-09-21
**Real-time classification, geolocation and interactive visualization of COVID-19 information shared on social media to better understand global developments**

**Rating:** 7
**Confidence:** 4

**Review:**

**Summary**
This work presented an interactive visualization system for COVID information including the number of affected people, prevention, treatment, death reports, etc. The system was based on Tweet streaming data with 4 COVID related filtering keywords. To classify tweets into different information categories (eg. number of affected people), the BERT-based pre-trained model was fined-tuned on the 2014 MERS dataset and the L2 layer was employed as an output layer. To further generate the information distributions on the global map, geolocations were extracted with name entity recognition from the text contents of tweets.

In this reviewer's view, this work is well written. This novel system provides a live information distribution that presents the changes in the pandemic situation.

**Comments**
1. To extract text-based geolocation, are tweets with misinformation filtered out? People may discuss things in other places, which may not true.

2. For the tweets that are not the first-person narrative, how do the maps of 'affected_people', 'deaths_reports' reflect the changes of the situation?

---

> ### Author Response · Authors · 2020-09-28
> **Response from the author**
>
> Thank you for the feedback.
>
> In response to your comments:
> 1.  Currently, there is no way for us to reliably filter misinformation automatically. When we built this tool, one of its intended use-cases was to assist the identification and characterization of misinformation by humans. In particular, user-specific information we capture, such as account age and number of followers can be useful in helping identify misinformation; however we chose not to display this information as it could lead to information overload.
> 2. Many tweets in these categories were of people sharing news articles, which can be useful for monitoring the spread of information/misinformation. However, being able to separate out first-person narratives would be valuable for many use-cases. While there is existing work on identifying eye-witness reports, we could not find labeled data for separating more general first-person narratives. This would be a very interesting avenue of future work.

---

### Official Review · AnonReviewer1 · 2020-09-25
**Good idea but lack of analysis and there is no evaluation**

**Rating:** 4
**Confidence:** 4

**Review:**

Even though the idea is interesting. But they are several issues that need to be addressed before considering it for publication.  for example  “training data” is not well described. The author mentioned that the annotated data is from Middle Eastern Respiratory Syndrome (MERS) outbreak but there is no clear connection with Covid-19 data, there are several Covid-19 datasets and the author did not mention why did not use them.

Main point: author did not report any classifier’s performance or any kind of evaluation metrics.  even though it is short paper but at least a table explaining the performance metrics and how the classifier performs. The classification is heart of this visualization and the performance metric s are must for any NLP or applied NLP research paper.

---

> ### Author Response · Authors · 2020-09-28
> **Response from the author**
>
> Thank you for the feedback.
>
> In response to your review:
>
> >“training data” is not well described
>
> We describe the training data in Section 2.2.1; with class descriptions and dataset statistics in Tables 1 and 2 respectively. For readers desiring further context on the training data, we cite the original paper with which the dataset was published.
>
> >there is no clear connection with Covid-19 data,
> >there are several Covid-19 datasets and the author did not mention why did not use them.
>
> Both MERS and COVID-19 are coronaviruses with similar symptoms.  However, its true that the MERS 2014 epidemic was nowhere near the current scale of the COVID-19 pandemic and that is a significant limitation of our training data. Nevertheless, when we originally built this dashboard at the end of January 2020, we were not aware of any labeled datasets of COVID-19 tweets. Furthermore, to the best of our knowledge, there are still no datasets with equivalent useful class labels.
>
> >author did not report any classifier’s performance or any kind of evaluation metrics
>
> We report the performance of our classifier at the end of Section 2.2.2; as well as per-class F1 scores in Table 3. In particular, we highlight that our novel L2 classification layer significantly outperforms traditional linear classification layers on rare classes  (where the inner product similarity used by linear classification layers has a bias against low-norm rare classes, as described in our paper and shown in Demeter et al., 2020).

---

### Official Review · AnonReviewer2 · 2020-09-28
**Critical information is needed**

**Rating:** 5
**Confidence:** 4

**Review:**

This paper proposes a real-time Twitter visualization tool which consists of three components: Twitter classification, geolocation extraction and interactive visualization.

I agree that the proposed system can be of great value in practical use. However, a lot of critical information is not included in this paper and without these information, it is hard to determine the contribution of this paper.

First, there exist a lot of twitter mapping/visualization works, e.g., real-time crisis mapping of natural disasters using social media and this one: https://blog.tensorflow.org/2019/09/disaster-watch-crisis-mapping-platform.html. All these tools have similar structure as that of the proposed tool: extract location of the twitter, perform classification on the twitter content and visualize the classified twitters in a map. The only difference to me is that the target labels in the classification layer is different. The author should include a comprehensive review of these visualization tools and highlight the differences between this paper and previous tools.

Second, a claim in this paper is that using negative L2 distances in the output classification layer outperforms using traditional inner product similarity. However, there is not enough experiments to support this claim. The marginal improvement in terms of validation accuracy (76.84 vs 76.35) is not convincing to me.

Finally, as the proposed system works in an interative and real-time manner, there should be an evaluation on the latency of the system.

---

> ### Author Response · Authors · 2020-09-28
> **Response from the author**
>
> Thank you for the valuable feedback.
>
> In response to your comments:
>
> >First, there exist a lot of twitter mapping/visualization works, e.g., real-time crisis mapping of natural disasters using social media and this one: https://blog.tensorflow.org/2019/09/disaster-watch-crisis-mapping-platform.html. All these tools have similar structure as that of the proposed tool: extract location of the twitter, perform classification on the twitter content and visualize the classified twitters in a map. The only difference to me is that the target labels in the classification layer is different. The author should include a comprehensive review of these visualization tools and highlight the differences between this paper and previous tools.
>
> Thank you for sharing previous work on real-time mapping of tweets; we were not aware of it, and will include a detailed review of these as well as a comparison with our method in the revised version of our paper. In particular, in contrast to the two references mentioned, we use neural named entity recognition and allow the user to differentiate between different types of geolocations (cities, countries, locations mentioned in user profile, locations mentioned in tweet text, and tweet geotags).
>
> > Second, a claim in this paper is that using negative L2 distances in the output classification layer outperforms using traditional inner product similarity. However, there is not enough experiments to support this claim. The marginal improvement in terms of validation accuracy (76.84 vs 76.35) is not convincing to me.
>
> In addition to accuracy, we show per-class F1 scores in Table 3 to highlight that our novel L2 classification layer significantly outperforms traditional linear classification layers on rare classes (where the inner product similarity used by linear classification layers has a bias against low-norm rare classes, as described in our paper and shown in Demeter et al., 2020). However, to further support our claim that our method performs better on rare classes, we can replicate this analyses on other datasets of crisis-related tweets and include the results in the revised version of our paper.
>
> >Finally, as the proposed system works in an interative and real-time manner, there should be an evaluation on the latency of the system.
>
> We update our dashboard every second and classify/geoparse tweets asynchronously. On a 2080Ti machine with an 8-core CPU, we were not able to process tweets as fast as they came in. However, removing retweets significantly alleviates the issue. We will include a more rigorous analysis of the system latency in the revised version of our paper.